# Relationship between Optimism, Self-Efficacy and Quality of Life: A Cross-Sectional Study in Elderly People with Knee Osteoarthritis

**DOI:** 10.3390/geriatrics8050101

**Published:** 2023-10-11

**Authors:** Agata Wojcieszek, Anna Kurowska, Anna Majda, Kinga Kołodziej, Henryk Liszka, Artur Gądek

**Affiliations:** 1Institute of Nursing and Midwifery, Faculty of Health Sciences, Jagiellonian University Medical College, 30-688 Krakow, Poland; 2Department of Orthopedics and Physiotherapy, Faculty of Health Sciences, Jagiellonian University Medical College, 30-688 Krakow, Poland

**Keywords:** osteoarthritis, self-efficacy, optimism, quality of life, gonarthrosis, old age

## Abstract

Background: Due to the presence of numerous problems in osteoarthritis, e.g., the presence of one or more chronic diseases, reduced self-esteem and reduced ability to cope, patients must undertake readaptation activities. In such circumstances, resources that are necessary for optimal adaptation become of particular importance. This cross-sectional study aimed to assess the impact of behavioral resources, namely self-efficacy and optimism, on quality of life perception in early-old-age patients with knee osteoarthritis. Methods: An anonymous survey was conducted using recognized research tools: the Index of Severity for Knee Disease, Life Orientation Test, General Self-Efficacy Scale and World Health Organization Quality of Life BEFF. The study involved 300 people aged between 60 and 75 years old, including 150 patients diagnosed with gonarthrosis and 150 people without diagnosed joint and muscular diseases of the lower limbs. Non-parametric tests (e.g., Mann–Whitney U test, Kruskal–Wallis test, Spearman’s correlation coefficient) were used for the statistical analysis of the results, assuming a significance level of *p* < 0.05. Results: The level of the examined personal resources was significantly lower in the group of people with gonarthrosis (*p* < 0.001), among whom low self-efficacy and a tendency toward pessimism prevailed. The results in terms of the level of lower limb joints impairment among the respondents correlated significantly and negatively with self-efficacy (r = −0.239; *p* = 0.003) and dispositional optimism (r = −0.318; *p* < 0.001). A higher level of the studied psychosocial resources led to a more favorable assessment of quality of life (*p* < 0.001) and own health (*p* < 0.001). In addition, a higher sense of self-competence was associated with better quality of life in the psychological (*p* = 0.044), social (*p* < 0.001) and environmental (*p* < 0.001) domains, while a tendency toward optimism was associated with higher quality of life perception in the social domain (*p* < 0.001). Conclusions: It would seem to be reasonable to introduce a routine diagnosis, assessing the level of personal capabilities of elderly people with knee osteoarthritis, which may have a beneficial effect on their perception of their quality of life and their own health.

## 1. Introduction

Osteoarthritis (OA) is the most common chronic disease of the musculoskeletal system. The multitude of risk factors for osteoarthritis suggests that its appearance is determined by a wide range of biological, mechanical and structural components [1]. The clinical manifestation of OA and the progression of the disease involve changes in the articular cartilage as well as the synovial membrane, subchondral bone, ligaments and muscles [2]. OA incidence increases with age and is relatively common, because for 35% of the population over 65 years old it concerns the knee joint (KOA; gonarthrosis). Four main symptoms predominate in KOA: pain, stiffness, decreased joint mobility and muscle weakness. All these symptoms can lead to quality of life impairment, with deterioration of the ability to perform daily activities [3].

Knee pain is of widespread nature and the reduction in physical functioning associated with it is a strong predictor of future disability. Many studies indicate that psychological factors have a large impact on the individual’s perception of pain [4,5]. From an analysis of scientific articles devoted to this subject, it can be concluded that there is a mutual relationship between pain severity and mental state. On the one hand, pain in OA can affect an individual’s mood and well-being [6,7]. On the other hand, a persistent negative cognitive style, characterized by helplessness, a tendency to exaggerate and reflective thoughts about one’s own pain, leads to its increased perception [8]. Its intensity may also be exacerbated by stress, lack of support and depression, which is more and more common in this group of patients (at a level of 40%) compared to the general population (at a level of 9%) [9,10,11]. Moreover, patients often feel pain during night rest, which negatively affects regenerative processes. Due to progressive mobility problems, the feeling of isolation increases [12]. Due to numerous problems occurring in osteoarthritis, patients must take reintegration measures. These are aimed at restoring mental well-being and maintaining social roles. In such circumstances, resources that are significant for optimal adaptation become of particular importance.

The concept of perceived self-efficacy was introduced by Albert Bandura. It is a view of the subject’s competencies, of how well it is equipped with the means to carry out intended actions. In other words, this component concerns the subject’s belief that he or she is capable of undertaking a specific activity or task, and of perseverance in achieving the intended goal in a specific situation [13]. Self-efficacy in the field of health can affect motivation and the decisions of subjects to take preventive action during illness [14], and, vice versa, improving the results of chronic diseases treatment is associated with improving self-efficacy [15]. Often, high self-efficacy is associated with healthy aging [16].

The concept of dispositional optimism was developed by Carver et al. [17]. According to these authors, this is a human trait expressed in generalized expectations regarding the results of one’s own actions. Its level determines the choice of goals and the effort put into their implementation. Moreover, it performs an autoregulatory function [17,18]. Available research results indicate that optimists report less pain than pessimists [19]. In another study conducted on a group of women with rheumatoid arthritis, it was found that greater optimism was associated with less catastrophizing about recurrent pain [20]. People differ in their tendency to be optimistic depending on many factors, such as current situation, life experiences and mental and emotional health. They may be optimistic in some circumstances, but more pessimistic or realistic in others.

Quality of life (QoL) is a rather imprecise and multidimensional concept that causes many definitional and methodological problems. This paper is based on a holistic construct derived from the definition of “health” introduced by the World Health Organization (WHO). The concept of quality of life understood in this way consists of somatic, psychological, social and environmental factors [21].

Recent clinical practice guidelines recommend a comprehensive approach to the management of osteoarthritis that includes identification, assessment, and management of general and pain-related psychological distress. Mood disorders such as depression and anxiety are familiar to many clinicians as distinct medical diagnoses and therefore may be detected more frequently in clinical practice. However, there are indications that less known or often overlooked constructs specific to pain, such as catastrophism, optimism or self-efficacy, should be assessed, which was the inspiration for designing this study [22,23].

This article is part of a series of thematically related scientific articles on the impact of physical and mental condition on the quality of life of patients with knee osteoarthritis. The main objective of this cross-sectional study was to present the levels of self-efficacy and dispositional optimism and to determine the relationship between these levels and the quality of life of subjects with symptomatic KOA. We hypothesized that self-efficacy and dispositional optimism would be negatively correlated with the level of disability of the lower limbs and might affect the perception of one’s own health and quality of life.

## 2. Materials and Methods

### 2.1. Research Design and Data Collection

This cross-sectional study was conducted from January 2019 to February 2020. The studies on the group of people with KOA (study group—SG) were carried out in orthopedic wards and clinics with the same profile. A request for permission to conduct a survey was sent to 7 health care facilities that provided advice on orthopedic surgery in Kraków. In the end, permission was obtained from 3 Kraków facilities.

The study on the comparative group (CG) was conducted in the period from May 2019 to November 2020 outside of health care facilities, i.e., the University of the Third Age and senior clubs in Kraków.

### 2.2. Participants

The study covered 174 people in the period of early old age (according to the WHO periodization of old age [24]), i.e., between 60 and 74 years old with diagnosed KOA (SG), and 170 people between 60 and 74 years old without diagnosed articular and muscular diseases of the lower limbs (CG). Due to the incomplete filling out of the prepared research tools, including failure to provide important socio-demographic data, 300 questionnaires (150 SG and 150 CG) were used for the analysis. The questionnaire return rate was 87.20%.

Patients from the study group were recruited on the basis of the following inclusion criteria: age between 60 and 74 years of age and diagnosis of KOA (determined by an orthopedic specialist using the American College of Rheumatology criteria). In addition, we assumed that patients with clinical manifestations of KOA would enter the study, so we enrolled those patients who had at least one of the following symptoms: pain, stiffness, or impaired knee function. Another criterion was the lack of surgical intervention related to the lower limbs within the year preceding the study and the absence of mental illnesses, i.e., depression and anxiety.

In the case of people from the comparative group, the following inclusion criteria were applied: age between 60 and 74 years old and no diagnosis of articular and muscular diseases of the lower limbs (OA, rheumatoid arthritis). As in the case of the SG, people in the CG could not have undergone a surgical intervention related to the lower limbs within the year preceding the study and could not have a diagnosis of mental illness, i.e., depression and anxiety.

### 2.3. Ethical Considerations

Prior to the commencement of the research, consents from hospital directors and the approval of the Bioethics Committee were obtained (KBET/1072.6120.1.2018). For the surveys, original and standardized research tools were used, which were selected in such a way that their content did not violate the interests of the study participants. When designing this study, the team of employees used the information contained in the Helsinki Declaration and adhered to the accepted rules of professional ethics. Participants in the study were provided with all the necessary information about the study and were informed that it was anonymous and that they could withdraw at any stage.

### 2.4. Instruments

The study used the estimation method, which enabled the collection of data on multi-stage scales, and the diagnostic survey method. The estimation method uses the estimation scale technique. The examined variables, i.e., self-efficacy, sense of optimism and quality of life were assessed using specific assessment criteria, i.e., degrees imposing a given order. In turn, as part of the diagnostic survey, the survey technique was used, and the research tools were an original survey questionnaire, which collected significant socio-demographic data (sex, age, place of residence, type of work performed in the past, declared duration of the disease) and anthropometric data (body weight, height), and the Lequesne pain–function index.

The Life Orientation Test (LOT-R) was used to measure the level of dispositional optimism. The authors of the Polish version of the questionnaire are Poprawa and Juczyński [25]. This tool contains 10 statements, 6 of which are diagnostic. The respondent answers the questions by marking a given number: 0—“definitely not related to me”, 1—“rather not related to me”, 2—“neither relevant nor irrelevant”, 3—“rather related to me” ”, 4—“definitely applies to me”. The overall test score ranges from 0 to 24 points and is the sum of six statements, including three positive (1, 4, 10) and three negative (3, 7, 9). The higher the score, the higher the level of dispositional optimism. The raw score can be recalculated according to the sten scale. Sten scores 1–4 indicate a low level of optimism (i.e., pessimism), 5–6 indicate a medium level (i.e., a neutral attitude) and 7–10 a high level (i.e., a tendency to be optimistic). In our study, the tool demonstrated satisfactory internal consistency, with a Cronbach alpha coefficient of 0.812.

The Generalized Self-Efficacy Scale (GSES) measures strength of general beliefs about the respondent’s effectiveness at coping with life’s problems and obstacles. The authors of the Polish version of the tool are Schwarzer, Jerusalem and Juczyński [25]. The respondent answers the questions by marking a given number: 1—“no”, 2—“rather not”, 3—“rather yes”, 4—“yes”. The overall score is the sum of all points obtained from 10 statements and ranges from 10 to 40 points. The higher the number, the higher the self-efficacy. When interpreting the result, the sten scale is used, where a sten score of 1–4 is a low result (10–24 points), 5–6 is an average result (25–29 points) and 7–10 is a high result (30–40 points). Self-efficacy is sometimes interchangeably expressed as a sense of self-competence. The Cronbach alpha coefficient was 0.84, proving the internal consistency of this tool.

The shortened WHOQOL-BREF Quality of Life Questionnaire was developed on the basis of the WHOQOL-100 questionnaire and is a universal research tool for assessing the quality of life of healthy and sick people. The Polish version of the tool was developed by Jaracz and Wołowicka from the Poznan University of Medical Sciences [26]. It contains 26 questions analyzing four areas of life: physical (question nos. 3, 4, 10, 15, 16, 17, 18), psychological (question nos. 5, 6, 7, 11, 19, 26), social (question nos. 20, 21, 22) and environmental (question nos. 8, 9, 12, 13, 14, 23, 24, 25). The score for each area of life is determined by calculating the arithmetic mean of the answers provided. The questionnaire also contains two questions analyzed separately, concerning individual perception of quality of life and one’s own health. The score ranges from 1 to 5 and has a positive direction—the higher the number of points, the better the quality of life. WHO approval was obtained for the use of the WHOQOL-BREF scale in this study, authorization number 371022 [27]. In our study, the Cronbach alpha coefficients were as follows: physical domain—0.72, psychological domain—0.87, social domain—0.62 and environmental domain—0.77.

The Lequesne pain and function index (ISK) is a questionnaire that assesses the functioning of the knee and hip joints. The Polish version of the questionnaire was made available by Nonna Anna Nowak, MD, PhD, from the Center for Osteoporosis and Osteo-Joint Diseases in Białystok [28,29]. The questionnaire consists of 11 questions. Five of them concern the perception of pain when walking, standing, rising from a sitting position or night rest, and morning discomfort. Each question is awarded from 0 to 2 points. One of these questions refers to the maximum distance covered and is assessed on a scale of 0 to 6 points, depending on the declared distance covered. The next four questions refer to activities related to everyday life, e.g., walking up the stairs, crouching or picking up an object from the floor, and are rated in the range of 0 to 2 points. The level of Cronbach’s alpha for this tool was 0.81.

### 2.5. Data Analysis

The study used statistical methods to develop and interpret the results. During the analysis of the research material, the analysis of quantitative variables was used, calculating the mean, standard deviation, median, quartiles, minimum and maximum. When comparing the values of quantitative variables in two groups, the Mann–Whitney U test was used. Correlations between quantitative variables were analyzed using Spearman’s rank correlation coefficient. The comparison of quantitative variables in three or more groups was performed by the use of the Kruskal–Wallis test. After detecting statistically significant differences, a post hoc analysis was performed using Dunn’s test to identify statistically significant groups. A significance level of *p* < 0.05 was adopted in all the tests performed. The research results were prepared using the statistical package R 4.0.1 [30].

## 3. Results

In both study groups there was a numerical predominance of women (54.67%). The mean age for the study group was 67 ± 4 years and for the comparative group was 66 ± 3 years. The majority of study group patients lived in rural areas (n = 83; 55.33%), while in the comparative group, a greater percentage of respondents lived in the city (n = 80; 53.33%). In both groups, white-collar work in the past was dominant (40.00%—SG; 46.00%—CG). In the case of KOA patients, a higher percentage of abnormal body weight was observed (overweight n = 69, i.e., 46.00%; obesity n = 4, i.e., 2.67%) compared to the comparative group (overweight n = 56, i.e., 37.33%; obesity n = 1, i.e., 0.67%).

Most of the surveyed patients with KOA felt pain in the knee joint during limb movement and in some positions during night rest (82.67%). In more than half of the patients with gonarthrosis, pain or discomfort persisted for more than a quarter of an hour (50.67%) after getting up from bed. Every second respondent reported that the symptoms were aggravated by standing for at least 30 min (64.00%), walking (68.00%) and standing up from a chair without using hands (69.34%). A dominant percentage of the examined patients could walk a maximum distance of about 1 km (39.33%), climbed stairs with little difficulty (47.33%) and used equipment to improve walking in the form of one elbow crutch (46.00%). In terms of everyday activities, going down the stairs caused little difficulty in the case of 34.67% of the respondents, although a similar result was also recorded for a significant level of difficulty—34.00%. Nearly half of the respondents found it difficult to crouch down (43.32%), and every second respondent found it difficult to move on uneven terrain (50.00%) (Table 1).

More than half of the SG patients presented low self-efficacy (58.67%) and a tendency to pessimism (55.33%). On the other hand, in the CG an average level of this resource (74.00%) and a neutral orientation (46.00%) dominated. The level of self-efficacy was significantly higher and life orientation more optimistic in the comparative group (*p* < 0.001) (Table 2).

The results in terms of the subjects’ level of impairment of the lower limb joints correlated significantly and negatively with the results in terms of their self-efficacy (r = −0.239; *p* = 0.003) and dispositional optimism (r = −0.318; *p* < 0.001) (Table 3).

The results obtained in terms of respondents’ self-efficacy correlated significantly and positively (r ˃ 0) with the results in the sphere of their perception of their own health (*p* < 0.001) and quality of life in the following domains: general (*p* < 0.001), psychological (*p* = 0.044), social (*p* < 0.001) and environmental (*p* < 0.001) (Table 4). The results in terms of respondents’ dispositional optimism correlated significantly and positively (r˃0) with the results in the sphere of their perception of their health (*p* < 0.001) and quality of life in the general (*p* < 0.001) and social (*p* < 0.001) domains (Table 4).

Age correlated significantly (*p* < 0.05) and negatively (r ˂ 0) with the perception of quality of life (r = −0.475, *p* < 0.001) and self-health (r = −0.495, *p* < 0.001), and with quality of life in the physical (r = −0.273, *p* = 0.001), psychological (r = −0.32, *p* < 0.001) and environmental (r = −0.242, *p* = 0.003) domains (Table 5).

The perception of quality of life (*p* = 0.018) and own health (*p* = 0.001) as well as psychological quality of life (*p* = 0.008) were significantly better in women. Quality of life in the physical (*p* < 0.001) and psychological (*p* < 0.001) domains was significantly better in the group of patients suffering up to a year or from 1 to 5 years than in the group of patients suffering from 6 to 10 years, whose quality of life in these domains was, in turn, significantly better than in the group suffering for more than 10 years. Perceptions of quality of life (*p* = 0.033) and own health (*p* < 0.001) were significantly better in the normal weight group (Table 6).

## 4. Discussion

The modern definition of health as biopsychosocial well-being requires that the patient’s condition be assessed in a broader context. Chronic disease management typically includes monitoring symptoms, adhering to treatment regimens, and keeping doctor appointments. The overall clinical changes that make up osteoarthritis can often be expressed in measurable terms. It is more difficult to assess mental and emotional burdens associated with the disease. Patients who experience chronic pain and activity limitation exhibit a complex range of psychosocial responses that are highly individualized. This fact gave us the premise that each disease entity, due to its specificity, deserves a separate assessment. What distinguishes this study from other studies is the restrictive selection of the study and comparative groups. By comparing people with symptomatic KOA and people with undisturbed motor function, we wanted to draw attention to the importance of mobility in older people. Moreover, in our opinion, the study is clinically valuable because it concerned the psychological determinants associated with the assessment of lower limb joint dysfunction. Information about life orientation and self-efficacy may draw attention to modifiable factors in order to improve the patient’s well-being and, consequently, the quality of life of older people with gonarthrosis.

As mentioned in the introduction, this is one of a series of articles devoted to the determinants of the quality of life of older people with knee osteoarthritis. In another scientific study, we presented and discussed in detail the research results obtained using, among other tools, the Lequesne index and the WOMAC scale (testing the level of lower limb disability). It should be mentioned that, in that study, among patients with osteoarthritis, the dominant group were people with severe (36.67%) and very severe (34.67%) lower limb joint function impairment and a high level of disability [31]. It is also worth noting that the whole range of complaints reported by the respondents in terms of pain, maximum walking distance and difficulties in performing daily activities showed a typical clinical picture of gonarthrosis. Statistical analysis of the results from this current study show that the level of personal capabilities possessed by respondents significantly differentiated both study groups, with a higher level recorded in the comparative group. The fact that the group of people with gonarthrosis was dominated by a low level of self-efficacy and a tendency to pessimism should be considered alarming. In this study, both examined variables were correlated, which meant that the greater the level of impairment of lower limb joint function, the lower the level of the examined behavioral resources. Meanwhile, the availability of various types of psychosocial resources may facilitate adaptation to perceived losses in late life [32], and constructive adaptive mechanisms may protect older adults from a cascade of poor health outcomes. In normalization studies conducted by Juczyński [25] on a group of 496 people aged 30–55, the highest level of self-efficacy was noted in women after mastectomy and people with diabetes, while the lowest level was diagnosed in dialysis patients or men after a heart attack. Similarly, in terms of life orientation, the highest index was observed in women after mastectomy and women with complicated pregnancies. The lowest level was observed in diabetics and menopausal women. Importantly, in the case of each disease entity mentioned above, higher values were obtained in terms of the examined personal resources than in our own research. This may result directly from the severity of KOA symptoms in the studied group of patients. It is worth mentioning that it is the knee joint (as the largest joint in the human body) that causes the most troublesome physical symptoms and the greatest limitations on the mobility of patients, and, in addition, the pain therapy used may cause further health problems (e.g., related to the use of non-steroidal anti-inflammatory medicines (NSAIDs)) and the waiting time for knee arthroplasty is several years in Poland. There are publications that emphasize the importance of personal resources, especially in diseases accompanied by pain. An observational study conducted by Degerstedt et al. [33] in a group of people with osteoarthritis of the knee and hip joints showed that patients with a high initial level of self-efficiency significantly more often reported pain reduction and increased physical activity after the therapy compared to patients with low levels of this resource. According to research conducted by Turner on a group of 140 retirees [34], people with a high level of self-efficacy were more likely to perceive pain-related problems as challenges to overcome and take effective ways to stop their impact. On the other hand, people with optimistic orientation may more often choose constructive coping strategies, focusing on planning or seeking support. Such an active attitude in the face of illness often improves health and pain outcomes, as patients who use this style of coping are more likely to engage in pro-health behaviors and seek social support [35]. There are scientific reports demonstrating that people who, in their considerations, tended to overestimate the impact of osteoarthritis on their functioning and emphasize its chronic duration had an increased risk of reporting more limitations than expected [36]. To sum up, a pessimistic attitude may result in the patient questioning various types of preventive actions [37]. There are many publications confirming that self-efficacy can be increased by learning disease management skills and self-control. In the case of patients with KOA, interventions will focus on effective pain management and increasing awareness of the disease itself [38]. It has also been shown that writing about and imagining a future in which everything has gone optimally increases optimism about the expectation of favorable outcomes, and this increase in optimism has been shown to be independent of mood [39].

Progress in diagnosis and treatment of many chronic diseases and the emergence of numerous initiatives promoting a healthy lifestyle result in extension of human life. The number of aging, old and elderly people is systematically increasing. However, apart from mere prolongation of the number of years lived, it is worth paying attention to the degree of broadly understood satisfaction with the totality of experiences taking place during this period. Our research showed the existence of a relationship between the level of personal resources and the quality of life. It has been proven that the higher the degree of the above-mentioned capabilities, the more value the respondents attributed to their life and health. In addition, a higher sense of self-competence was associated with a better quality of life in the psychological, social and societal domains, while a higher level of life orientation was associated with a higher perception of quality of life in the social domain. It is extremely interesting that the possession of the personal resources was not related to the assessment of the physical domain, which consisted of the following aspects: daily activities, addiction to medicinal substances and auxiliary substances, energy and fatigue, mobility, pain and discomfort, sleep and rest, and ability to work. This may mean that the resources themselves did not affect direct assessment of physical performance as broadly understood. Nevertheless, the positive impact of possession of personal resources on quality of life has been confirmed in Polish [40] and also foreign publications [41,42,43]. In our research, better perception of one’s own health and quality of life was related to younger respondent age, female gender, shorter disease duration and body weight within the normal range. Overall, women would be expected to have a poorer quality of life due to the fact that female OA tends to be more severe than the male disorder. However, in the case of illness, women are more likely to seek, e.g., social support, which may significantly improve their QoL scores.

This study has several limitations. The first is related to the design of the cross-sectional study itself, which makes it impossible to assess causality. In the course of the study, no data on the occurrence of comorbidities and ways of treating them in the studied groups of patients were obtained, which would seem to be of great importance for many indicators assessed in this study. The methods used to obtain the results took the form of a questionnaire. In addition to questionnaires, there are other objective and more sensitive tests, such as checking some mental, functional or combined characteristics. Also, no information was obtained on the analgesic therapy used in the course of KOA, which could have influenced, for example, the results obtained in terms of the level of disability of the lower limbs. In addition, the WHOQOL-BREF scale used could be replaced with the WHOQOL-OLD scale, which applies strictly to the elderly. It should be remembered that when evaluating the quality of life in a group of elderly people, one should often signal that the subject of the assessment is their present situation, as there is a noticeable tendency to look retrospectively at the achievements of previous years and, thus, to make a purely emotional assessment.

The results obtained in this study may constitute a reason for further in-depth research, taking into account the aspects mentioned in the limitations. It would be very valuable to introduce a specific psychological intervention to improve psychosocial resources and observe the potential effects. Moreover, it is worth conducting research with a comparative group consisting of people with a similar disease, e.g., rheumatoid arthritis.

## 5. Conclusions

A higher level of the studied psychosocial resources was associated with better perception of quality of life and own health in the group of people in early old age with KOA. The study showed that the level of self-efficacy was significantly higher and the life orientation more optimistic in the group of elderly people with undisturbed motor skills. At the same time, the studied patients with gonarthrosis presented low self-efficacy and a tendency toward pessimism.In view of the obtained results, it would seem reasonable to introduce a routine diagnosis of levels of personal capabilities possessed, especially in the group of chronically ill people, which may have a positive impact not only on indicators related to the disease, but also on broadly understood quality of life. Promoting screening tests in this area, combined with education on successful aging in the gradually growing global elderly population, could bring tangible results.

## Figures and Tables

**Table 1 geriatrics-08-00101-t001:** Scores of KOA patients studied in terms of pain, maximum distance and difficulty in performing daily activities.

SG		
Lequesne Index—Category Pain	**Points**	
	**n**	**%**
1. Do you experience pain in your knee joint during night rest?		
(a) No	1	0.67
(b) Yes, only with movements or in certain positions	124	82.67
(c) Yes, even without movement	25	16.66
2. Do you experience pain or discomfort in your knee joint in the morning after getting out of bed?		
(a) Lasting up to one minute	0	0
(b) Yes, but no longer than 15 min	74	49.33
(c) Yes, longer than 15 min	76	50.67
3. Do you experience difficulty or increased pain if you have to stand in one place for 30 min?		
(a) No	54	36.00
(b) Yes	96	64.00
4. Do you experience pain or discomfort in your knee joint while walking?		
(a) No	24	16.00
(b) Yes, but it only occurs after a certain distance has been traveled	102	68.00
(c) Yes, immediately after starting from a standstill, and it intensifies while walking	24	16.00
5. Do you have difficulty or increased pain in the knee joint when standing up from a chair without using your hands?		
(a) No	46	30.66
(b) Yes	104	69.34
**Lequesne Index—Category** **Maximum walking distance**	**Points**	
	**N**	**%**
6. How long a distance can you cover?		
(a) Unlimited	0	0
(b) Above 1 km	37	24.67
(c) About 1 km (in 15 min)	59	39.33
(d) From 500 to 900 m (about 8–15 min)	30	20.00
(e) From 300 to 500 m	7	4.66
(f) From 100 to 300 m	16	10.67
(g) Less than 100 m	1	0.67
7. Do you need any movement assistance?		
(a) No	58	38.66
(b) Yes, one cane or crutch	69	46.00
(c) Yes, two canes or crutches	23	15.34
8. Can you walk up the stairs to the top?		
(a) Yes, without difficulty	4	2.67
(b) Yes, with little difficulty	71	47.33
(c) Yes, but with difficulty	49	32.67
(d) Yes, with great difficulty	26	17.33
(e) No, it is impossible	0	0
**Lequesne Index—Category** **Daily activities**	**Points**	
	**N**	**%**
9. Can you go down stairs?		
(a) Yes, easily	0	0
(b) Yes, with little difficulty	52	34.67
(c) Yes, with difficulty	47	31.33
(d) Yes, with great difficulty	51	34.00
(e) No, it is impossible	0	0
10. Can you kneel or crouch?		
(a) Yes, easily	2	1.36
(b) Yes, with little difficulty	56	37.32
(c) Yes, but with difficulty	65	43.32
(d) Yes, with great difficulty	27	18.00
(e) No, it is impossible	0	0
11. Can you walk on uneven ground?		
(a) Yes, easily	0	0
(b) Yes, with little difficulty	39	26.00
(c) Yes, but with difficulty	75	50.00
(d) Yes, with great difficulty	36	24.00
(e) No, it is impossible	0	0

n—number, SG—study group. Source: Own study.

**Table 2 geriatrics-08-00101-t002:** Level of self-efficacy and life orientation in SG and CG and comparison in terms of self-efficacy and dispositional optimism.

Overall Self-Efficacy Index [Categories]		SG		CG		Total	
		n	%	n	%	n	%
Low		88	58.67	25	16.67	113	37.67
Average		40	26.67	111	74.00	151	50.33
High		22	14.67	14	9.33	36	12.00
**Level of dispositional** **optimism** ** [categories]**		**SG**		**CG**		**Total**	
		n	%	n	%	n	%
A tendency to pessimism		83	55.33	31	20.67	114	38.00
Neutral orientation		43	28.67	69	46.00	112	37.33
Tendency to optimism		24	16.00	50	33.33	74	24.67
**Overall self-efficacy index**			**SG**		**CG**		** *p* ** *****
		**(N = 150)**			**(N = 150)**		
[points]	M ± SD	23.42 ± 4.65			26.61 ± 3.54		*p* < 0.001
	Me	24			27		
	Q1–Q3	19–27			25–29		
**Level of dispositional optimism**		**SG**			**CG**		** *p* ** *****
		**(N = 150)**			**(N = 150)**		
[points]	M ± SD	11.62 ± 4.46			14.67 ± 2.84		*p* < 0.001
	Me	9.5			15		
	Q1–Q3	8–15			13–17		

M—mean, SD—standard deviation, Me—median, Q1—1st quartile, Q3—3rd quartile, *p*—significance level, SG—study group, CG—comparative group. * Mann–Whitney U test. Source: Own study.

**Table 3 geriatrics-08-00101-t003:** Relationship between the level of lower limb disability and the level of dispositional optimism and self-efficacy.

Level of Impairment of the Lower Limbs Functions	Correlation with the Level of Self-Efficacy	Correlation with the Level of Optimism
	Spearman’s Correlation Coefficient	
Total score	r = −0.239; *p* = 0.003	r = −0.318; *p* < 0.001

*p*—statistical value, r—Spearman’s rank correlation coefficient. Source: Own study.

**Table 4 geriatrics-08-00101-t004:** Relationship between respondents’ perception of their quality of life and their sense of self-efficacy and dispositional optimism.

WHOQOL-BREF [Points]	Correlation with the Level of Self-Efficacy	Correlation with the Level of Optimism
	Spearman’s Correlation Coefficient	
Overall quality of life	r = 0.576; *p* < 0.001	r = 0.523; *p* < 0.001
Satisfaction with health	r = 0.43; *p* < 0.001	r = 0.647; *p* < 0.001
Physical domain	r = −0.038; *p* = 0.642	r = −0.009; *p* = 0.915
Psychological domain	r = 0.164; *p* = 0.044	r = 0.059; *p* = 0.475
Social domain	r = 0.369; *p* < 0.001	r = 0.592; *p* < 0.001
Environmental domain	r = 0.289; *p* < 0.001	r = 0.099; *p* = 0.228

*p*—statistical value, r—Spearman’s rank correlation coefficient. Source: Own study.

**Table 5 geriatrics-08-00101-t005:** Relationship between the age of the patients with KOA and their perception of their own health and quality of life.

WHOQOL-BREF [Points]	Correlation with Age
	Spearman’s Correlation Coefficient
Overall quality of life	r = −0.475, *p* < 0.001
Satisfaction with health	r = −0.495, *p* < 0.001
Physical domain	r = −0.273, *p* = 0.001
Psychological domain	r = −0.32, *p* < 0.001
Social domain	r = −0.082, *p* = 0.316
Environmental domain	r = −0.242, *p* = 0.003

*p*—statistical value, r—Spearman’s rank correlation coefficient. Source: Own study.

**Table 6 geriatrics-08-00101-t006:** Relationship between gender, duration of the disease and BMI, and perception of own health and quality of life.

WHOQOL-BREF [Points]			Gender			*p* *
			Female (N = 82)		Male (N = 68)	
Overall quality of life		M ± SD	3.33 ± 0.67		3.06 ± 0.93	*p* = 0.018
		Me	3		3	
		Q1–Q3	3–4		2–4	
Satisfaction with health		M ± SD	3.07 ± 0.54		2.65 ± 0.77	*p* < 0.001
		Me	3		3	
		Q1–Q3	3–3		2–3	
Physical domain		M ± SD	11.63 ± 1.58		11.32 ± 1.99	*p* = 0.069
		Me	11		11	
		Q1–Q3	11–13		10–12.25	
Psychological domain		M ± SD	13.05 ± 2.01		12.49 ± 1.88	*p* = 0.008
		Me	13.5		12	
		Q1–Q3	12–14		11–13.25	
Social domain		M ± SD	12.26 ± 1.58		12.62 ± 1.74	*p* = 0.248
		Me	12		12	
		Q1–Q3	12–13		12–13	
Environmental domain		M ± SD	12.33 ± 1.51		12.74 ± 1.64	*p* = 0.552
		Me	12		12	
		Q1–Q3	12–13		12–14	
**WHOQOL-BREF** [points]		**Duration of the disease**				***p* ****
		Up to 1 year A (N = 11)	1–5 years B (N = 63)	6–10 years C (N = 39)	Over 10 years D (N = 37)	
Overall quality of life	M ± SD	4 ± 0.63	3.24 ± 0.53	3.38 ± 0.85	2.73 ± 0.93	*p* < 0.001 A > C,B > D
	Me	4	3	3	2	
	Q1–Q3	4–4	3–3	3–4	2–4	
Satisfaction with health	M ± SD	3.27 ± 0.65	3 ± 0.57	2.87 ± 0.7	2.57 ± 0.77	*p* = 0.001 A,B,C > D
	Me	3	3	3	2	
	Q1–Q3	3–3	3–3	3–3	2–3	
Physical domain	M ± SD	13.09 ± 1.45	12.17 ± 1.24	11.33 ± 2.25	10.03 ± 0.8	*p* < 0.001 A,B > C > D
	Me	13	12	11	10	
	Q1–Q3	13–13	11–13	10–12.5	9–11	
Psychological domain	M ± SD	14.18 ± 1.17	13.76 ± 1.27	12.41 ± 2.35	11.14 ± 1.34	*p* < 0.001 A,B > C > D
	Me	14	14	13	11	
	Q1–Q3	13.5–14.5	13–15	11–14	11–12	
Social domain	M ± SD	12 ± 1.79	12.35 ± 0.7	12.36 ± 2.53	12.73 ± 1.64	*p* = 0.144
	Me	11	12	12	12	
	Q1–Q3	11–12	12–13	12–15	12–15	
Environmental domain	M ± SD	13.45 ± 1.57	12.73 ± 1.41	12.59 ± 2.2	11.78 ± 0.42	*p* < 0.001 A,B,C > D
	Me	13	12	12	12	
	Q1–Q3	13–13	12–14	12–14	12–12	
**WHOQOL-BREF** [points]		**BMI**				***p* ***
		Normal (N = 77)		Overweight or obesity (N = 73)		
Overall quality of life	M ± SD	3.32 ± 0.8		3.08 ± 0.8		*p* = 0.033
	Me	3		3		
	Q1–Q3	3–4		3–3		
Satisfaction with health	M ± SD	3.08 ± 0.62		2.67 ± 0.69		*p* < 0.001
	Me	3		3		
	Q1–Q3	3–3		2–3		
Physical domain	M ± SD	11.58 ± 1.8		11.4 ± 1.75		*p* = 0.386
	Me	11		11		
	Q1–Q3	11–13		10–13		
Psychological domain	M ± SD	12.71 ± 2.01		12.88 ± 1.92		*p* = 0.59
	Me	13		13		
	Q1–Q3	11–14		12–14		
Social domain	M ± SD	12.4 ± 1.73		12.44 ± 1.58		*p* = 0.21
	Me	12		12		
	Q1–Q3	11–13		12–13		
Environmental domain	M ± SD	12.32 ± 1.69		12.71 ± 1.43		*p* = 0.094
	Me	12		12		
	Q1–Q3	12–13		12–14		

M—mean, SD—standard deviation, Me—median, Q1—1st quartile, Q3—3rd quartile. *p*—significance level, BMI—body mass index. * Kruskal–Wallis test. ** Kruskal–Wallis test + post hoc analysis (Dunn’s test).

## Data Availability

The data presented in this study are available on request from the corresponding author. The data are not publicly available due to the privacy requirements of participants.

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
