# Peer review of "Relationship between Optimism, Self-Efficacy and Quality of Life: A Cross-Sectional Study in Elderly People with Knee Osteoarthritis"

_geriatrics, 2023, doi:10.3390/geriatrics8050101_

Round 1

Reviewer 1 Report

The authors have carried out an interesting study that leads to conclusions that are consistent with other series mentioned in the introduction to their paper, regarding the association of low mood and depression with osteoarthritis pain. The study was well conducted and the ethical postulates of the study were adequately addressed, which is not always the case in this type of studies.

On the one hand, while reading, I noticed some shortcomings, some of which were later mentioned by the authors in the discussion: the presence of comorbidities, the use of concomitant medication for these comorbidities, the use of anxiolytics, treatments received for osteoarthritis beyond traditional analgesics, etc. were not controlled for. It would also have been interesting to determine the degree of osteoarthritis in the study group: this last gap should be mentioned in the discussion.

In my opinion, the article suffers from an excess of information that could be simplified without affecting the essence of the study. For instance:

Materials and Methods

-      Table 1: This is a very exhaustive table. If possible, it could be simplified by giving a mean result for each domain (pain, walking distance and daily activities ) obtained from the score of each patient for each of the domains, according to Lequesne. This procedure, will allow to know the overall situation of the Study Group and also how many patients are in each degree of severity (mild, moderate, severe, extremely severe).

-      Consider to unify Tables 2 and 3.

-      Table 5 can be simplified in the form:

WHOQOL-BREF

Level of self-efficacy

Level of optimism

Discussion

I suggest to reduce the discussion. It is too extensive and repetitive. In particular, the second paragraph is too long. If the authors consider of relevance all the references, they can maintain them but summarizing the information.

Minor issues:

-      Sometimes the study group is named as GS instead of SG.

-      Please review the heading of Table 7: I think it is not correct.

-   

Please have the publication reviewed by a native English speaker.

Reviewer 2 Report

This cross-sectional study aimed to assess the impact of behavioral resources, specifically self-efficacy and optimism, on the quality of life perception among early old-age patients with knee osteoarthritis. A survey involving 300 participants aged 60 to 75 utilized established research instruments, including the Index of Severity for Knee Disease, Life Orientation Test, General Self-Efficacy Scale, and World Health Organization Quality of Life BEFF. Non-parametric tests were applied for statistical analysis, revealing that individuals with gonarthrosis exhibited significantly lower levels of self-efficacy and higher pessimism. Impairment in lower limb joints correlated negatively with self-efficacy and dispositional optimism. Higher psychosocial resource levels were associated with improved quality of life and self-perceived health, particularly in psychological, social, and environmental domains. Consequently, routine assessments of personal potentials in elderly knee osteoarthritis patients may enhance their quality of life and health perception.

The work is interesting and overall well-written. The introduction is clear and well-referenced. What is lacking is a greater focus on conveying the novelty of the study. The issue at hand is: what should differentiate gonarthrosis in causing individuals suffering from it to have a more pessimistic attitude and lower satisfaction levels compared to other conditions (assuming there are differences)? The impression is that (although well-executed), the study replicates other studies focused on different conditions. The authors should better emphasize this aspect in the Introduction and Discussion. The Methods are well-described. I would ask the authors to check the acronyms (are they all necessary?). Stylistically, I would place the Ethical considerations just after the Participants section. The results are clear. Just one point: why are some significances highlighted with an asterisk while others are not? I would ask the authors to clarify this aspect further. The Discussion is well-written, with the exception of what was mentioned in the general comment.

English is fine with only some minor editing required.

Round 2

Reviewer 1 Report

The answers and the work done by the authors to improve the paper is satisfactory. 

Reviewer 2 Report

The Authors replied adequately to all my previous suggestions. No further revisions are required.